# Curvature-Induced Membrane Remodeling by the Cell-Penetrating Peptide Pep-1

**DOI:** 10.3390/membranes15120373

**Published:** 2025-12-03

**Authors:** Yasith Indigahawela Gamage, Jianjun Pan

**Affiliations:** Department of Physics, University of South Florida, Tampa, FL 33620, USA; yasith@usf.edu

**Keywords:** cell-penetrating peptide, atomic force microscopy, membrane remodeling, electrostatic and hydrophobic interactions, membrane curvature

## Abstract

The cell-penetrating peptide Pep-1 interacts with lipid membranes through combined electrostatic and hydrophobic forces, yet the structural details of its membrane remodeling activity remain unclear. Using atomic force microscopy (AFM), we examined how Pep-1 perturbs supported lipid bilayers of varying composition and geometry. In zwitterionic POPC bilayer patches, Pep-1 preferentially targeted patch boundaries, where lipid packing is less constrained, leading to edge erosion and detergent-like disintegration. Incorporation of anionic POPS enhanced peptide binding and localized disruption, giving rise to elevated annular rims, holes, and peptide–lipid aggregates. In cholesterol-containing POPC bilayer patches, Pep-1 induced extensive surface reorganization marked by protruded, ridge-like features, consistent with lipid redistribution and curvature generation. In continuous POPC/POPS bilayers lacking free edges, Pep-1 formed discrete, flower-like protrusions that coalesced into an interconnected network of thickened peptide-rich domains. These findings reveal composition-dependent remodeling pathways in which Pep-1 destabilizes, reorganizes, or curves membranes according to their mechanical and electrostatic properties, providing new insight into peptide–membrane interactions relevant to cell-penetrating peptide translocation.

## 1. Introduction

Cellular and tissue barriers limit the intracellular delivery of many therapeutics. One promising approach to overcome this limitation involves the use of cell-penetrating peptides (CPPs)—short peptides capable of traversing biological membranes and transporting bioactive cargoes. The discovery of the HIV-1 TAT peptide [1,2] and the Drosophila Antennapedia-derived penetratin [3,4] established the foundation for this field, leading to the identification of numerous natural and synthetic CPPs.

In recent years, the CPP field has rapidly expanded, driven by advances in peptide design and intracellular delivery technologies. Modern CPP engineering efforts focus on improving cargo versatility, enhancing delivery efficiency, and expanding applicability to primary cells and in vivo systems. For example, the amphiphilic RALA peptide has emerged as a robust delivery vector for CRISPR gene-editing cargos of multiple formats [5]. Other work has explored CPP-functionalization of nanocarriers to promote receptor-specific endocytosis and enhance transdermal penetration [6]. CPPs have also been incorporated into peptide–drug conjugates to overcome therapeutic resistance in cancer, where membrane-active carriers improve cytosolic delivery and selectively kill drug-resistant melanoma cells while sparing healthy cells [7]. Complementary studies have optimized CPPs for delivery of antisense peptide nucleic acids (PNAs), demonstrating enhanced nuclear localization and efficient gene-targeting activity in vitro [8]. In parallel, structure-guided CPP engineering has produced highly permeant CPPs capable of delivering proteins into mammalian cells [9]. Collectively, these recent developments underscore the continued relevance of CPPs as highly adaptable molecular transporters and highlight ongoing efforts to tune peptide chemistry for optimized membrane interaction, intracellular delivery, and therapeutic potential.

Although CPPs and antimicrobial peptides (AMPs) are often regarded as distinct classes, accumulating evidence shows that their membrane activities partially overlap [10,11]. Several well-studied CPPs also display antimicrobial activity at micromolar concentrations, while still entering cells without causing overt membrane lysis [12]. In microbes, such peptides can traverse the membrane through nonendocytic, energy-independent pathways, often without catastrophic membrane failure. Conversely, some AMPs are able to penetrate membranes or access cytoplasmic components without membrane rupture. Thus, CPPs and AMPs share key physicochemical features—cationic charge, amphipathicity, and the ability to transiently reorganize lipid packing—while differing primarily in biological specificity and the concentration range at which disruptive effects emerge. This overlap highlights the importance of understanding membrane perturbation as a continuum, rather than a strict dichotomy between “non-lytic” CPPs and “lytic” AMPs.

Pep-1 is a synthetic amphipathic CPP designed for non-covalent delivery of macromolecules [13]. Its sequence contains a hydrophobic tryptophan-rich segment, a proline spacer, and a lysine-rich cationic domain. Pep-1 is stable under physiological conditions and exhibits strong affinity for both neutral and anionic membranes [14]. However, despite extensive use as a delivery vector, the molecular mechanism of Pep-1–induced membrane translocation remains unclear.

The internalization of CPPs is generally understood to proceed through two broad pathways: endocytosis and direct membrane translocation. The latter refers to energy-independent entry processes that bypass classical endocytic routes and can involve a variety of mechanisms, such as inverted micelle formation [4], transient pore formation [15,16], curvature-driven deformation of the lipid matrix [17,18], electroporation-like permeabilization [19], and fusion or leaky fusion events [20,21]. Additional models emphasize peptide clustering and membrane destabilization, including localized thinning, carpet-like adsorption, and the sinking-raft mechanism [22,23,24]. A concentration-dependent dual mechanism has also been proposed, wherein direct translocation predominates at low peptide levels, while endocytic uptake becomes more prominent as peptide concentration increases [25]. The dominant pathway for any given CPP depends on multiple factors, including peptide sequence, charge distribution, lipid composition, and experimental conditions [26].

Experimental findings on Pep-1 have shown both energy-dependent and independent uptake. Pep-1 can rapidly enter cells under standard culture conditions [27], and uptake occurs even at 4 °C, consistent with direct translocation. However, fluorescent sterol labeling suggests an additional endocytic component [28]. Circular dichroism (CD) spectroscopy revealed that Pep-1 transitions from a random coil in aqueous solution to an α-helical structure in lipid environments [29,30]. Sum frequency generation spectroscopy further showed that Pep-1 adopts β-sheet structures in gel-phase DPPG and mixed α-helical/β-sheet structures in fluid-phase POPG bilayers [31]. Mechanistic studies proposed both barrel-stave pore formation [14] and surface adsorption-induced membrane destabilization [30], emphasizing the need for nanoscale structural characterization.

In this work, we used atomic force microscopy (AFM) to directly visualize how Pep-1 interacts with supported lipid bilayers of varying composition and geometry. To capture both central and edge-specific effects, we employed lipid bilayer patches—discrete, island-like domains. These patches, typically hundreds of nanometers to micrometers in diameter, provide distinct central and boundary regions separated by exposed mica. Unlike continuous supported bilayers that offer uniform topography, patch systems enable simultaneous observation of peptide effects at edges and interiors, revealing early remodeling events. Similar AFM-based studies have been reported for other CPPs [32] and antimicrobial peptides [33]. Because Pep-1 engages membranes through both electrostatic and hydrophobic interactions, the bilayer patch platform offers a useful means to resolve how peptide binding initiates at edges, propagates into membrane interiors, and promotes curvature generation and structural reorganization relevant to peptide translocation.

## 2. Materials and Methods

### 2.1. Materials

1-palmitoyl-2-oleoyl-*sn*-glycero-3-phosphocholine (POPC), 1-palmitoyl-2-oleoyl-*sn*-glycero-3-phospho-L-serine (POPS), and cholesterol were purchased as lyophilized powders from Avanti Polar Lipids (Alabaster, AL, USA). Pep-1 peptide was obtained in powder form from MedChemExpress (Monmouth, NJ, USA) with a reported purity of 99.27%. The peptide sequence is **K** *E* T **W W**
*E* T **W W** T *E* **W** S Q P **K K K R K V**, where positively charged residues (K, R) are shown in bold, negatively charged residues (E) in italics, and hydrophobic residues (W, V) are bold and underlined.

### 2.2. Preparation of Small Unilamellar Vesicles (SUVs)

Lipid stock solutions were prepared by dissolving individual lipid powders in chloroform. Desired lipid compositions were obtained by mixing appropriate volumes of these stock solutions in glass tubes. The organic solvent was first removed under a gentle nitrogen stream using a 12-position N-EVAO evaporator (Organomation Associates, Berlin, MA, USA), followed by vacuum desiccation for at least one hour to eliminate residual solvent. The resulting dry lipid films were rehydrated with Milli-Q ultrapure water. SUVs were generated by ultrasonication using a Branson Sonifier SFX 250 (Brookfield, CT, USA) equipped with a flat-tip probe, while maintaining the samples in an ice-water bath to prevent overheating. The vesicle suspensions were centrifuged briefly to remove any particulates. The average SUV radius was about 30–50 nm, depending on lipid composition, as determined by dynamic light scattering (Dynapro Nanostar, Wyatt Technology, Santa Barbara, CA, USA).

### 2.3. AFM Sample Preparation

AFM experiments at room temperature (~23 °C) were carried out using a Multimode 8 Atomic Force Microscope equipped with a Nanoscope V controller (Bruker, Santa Barbara, CA, USA). The procedures followed our previously described protocols [34,35,36]. Lipid SUVs were diluted in 5 mM CaCl_2_ to a final concentration of 15–20 µg/mL for patch formation or 60 µg/mL for continuous bilayer formation. The presence of Ca^2+^ ions facilitates vesicle fusion on hydrophilic mica. The diluted vesicle solution was introduced into a liquid AFM cell containing a freshly cleaved mica substrate pre-equilibrated with 5 mM CaCl_2_. After an incubation period of ~5 min to allow vesicle rupture and bilayer formation, the liquid cell was gently flushed with a buffer containing 10 mM HEPES (pH 7.4) and 2 mM CaCl_2_ (hereafter referred to as HEPES buffer) to remove unfused vesicles.

### 2.4. AFM Imaging and Analysis

All imaging was performed in liquid using PeakForce Quantitative Nanomechanical (QNM) mode at a scan rate of 1 Hz and a resolution of 256 pixels per line. Silicon nitride probes (Bruker DNP-S10, Fällanden, Switzerland, tip A; nominal tip radius 10 nm; spring constant ~0.3 N/m) were used. The peak force setpoint was optimized for each experiment to minimize perturbation of the bilayer surface. After initial imaging of the supported bilayer, Pep-1 was introduced into the AFM liquid cell in a stepwise manner using a syringe pump to achieve sequential increases in peptide concentration. The probe position was maintained over the same region to monitor concentration-dependent structural transitions; however, minor lateral drift between consecutive scans was unavoidable.

For each experimental condition, AFM measurements were performed in at least two independent trials. The obtained AFM height data were processed by first-order polynomial flattening to remove background curvature. Image analyses and quantitative measurements were performed using custom Python scripts.

## 3. Results and Discussion

### 3.1. POPC Bilayer Patches

Initial investigations were conducted on zwitterionic POPC bilayer patches exposed to increasing concentrations of Pep-1 (0, 1, 2, and 4 µM). As shown in Figure 1, the control bilayer patches displayed smooth surfaces. After introducing 1 µM Pep-1 into the AFM liquid cell, pronounced morphological changes were observed. The large, continuous bilayer patches present in the control began to disintegrate. Pep-1 appeared to interact preferentially with the patch boundaries, promoting edge disruption and the formation of numerous small fragments. Many of these fragments exhibited worm-like morphologies. Although large portions of the bilayer interior remained intact, localized perturbations and surface defects were evident across all patches.

At 2 µM Pep-1, bilayer disruption became more extensive. The residual worm-like features observed at 1 µM were further fragmented into smaller, granular structures, resulting in highly irregular patch boundaries. Within the patch interiors, the initial surface perturbations evolved into nanoscale defects, producing a roughened texture. The overall morphology suggests that the bilayer patches transformed into a percolating network of interconnected lipid clusters, indicative of progressive structural collapse and loss of bilayer continuity.

At 4 µM Pep-1, the degree of disruption was comparable to that observed at 2 µM, suggesting that the system had reached a steady state of peptide-induced destabilization. High-resolution imaging over a 500 nm × 500 nm area (Figure 1E) revealed that the patch boundaries had largely dissolved into small, particle- or disk-like structures, many of which remained loosely connected to the main patch. These features likely correspond to micellar or discoidal peptide–lipid complexes, characteristic of detergent-like solubilization [37]. The bilayer interior exhibited shallow nanoscale pits (~1 nm deep), consistent with localized thinning. Similar effects—characterized by an increased area per lipid and a corresponding reduction in bilayer thickness—have been reported for other CPPs interacting with lipid bilayers [38].

These structural signatures collectively support the interpretation that a detergent-like lysis mechanism [39] underlies the Pep-1–induced disruption of POPC bilayer patches. This mechanism arises from the amphipathic nature of Pep-1: its tryptophan-rich hydrophobic segment inserts into the bilayer interface, where indole groups interact with the lipid acyl–carbonyl region through hydrophobic and hydrogen-bonding interactions, while the positively charged lysine/arginine-rich domain associates with zwitterionic headgroups. The balance of these hydrophobic and electrostatic interactions drives partial insertion, lipid extraction, and the formation of mixed peptide–lipid aggregates. The strength of this interaction is further modulated by the bilayer’s interfacial electrostatic and dipole potentials [40].

Because lipid packing is less constrained at patch boundaries, which exhibit reduced line tension [41], hydrophobic insertion of Pep-1 is energetically favored at these sites. Consequently, membrane disruption is initiated preferentially along the edges [42], where the peptide inserts and accumulates. This insertion perturbs local lipid organization and generates lateral stress, promoting lipid extraction and the formation of peptide–lipid complexes. Similar edge destabilization has also been reported for the antimicrobial peptide PG-1 interacting with DMPC bilayer patches [33]. As disruption progresses, the bilayer disintegrates into smaller fragments, which markedly increases the total boundary length relative to the initial patch. The expansion of available edges further facilitates peptide insertion and lipid extraction, establishing a positive feedback loop that accelerates membrane disintegration. Ultimately, this self-propagating process leads to detergent-like dissolution of the bilayer into micellar or discoidal structures.

Beyond the boundary regions, Pep-1 also induces shallow defects within the patch interiors. These defects likely arise from localized peptide binding and accumulation, which promote local lipid extraction or rearrangement. However, the more tightly packed lipid environment within the patch interior confers greater structural stability, rendering it less susceptible to Pep-1–induced disruption compared to the boundary.

Overall, these findings indicate that Pep-1 disrupts POPC bilayer patches through a boundary-initiated, detergent-like mechanism in which peptide insertion and lipid extraction progressively compromise bilayer integrity.

### 3.2. Effect of Anionic POPS

To examine the effect of anionic lipids, POPC + 30 mol% POPS bilayer patches were analyzed (Figure 2). The untreated patches exhibited a few small pits distributed across their surfaces. These pits likely arise from incomplete bilayer formation during sample preparation.

Upon addition of 1 µM Pep-1, pronounced structural changes were observed. Large, discrete holes began to develop within the patch interiors, often originating from the preexisting pits. Each hole was typically surrounded by an elevated annular rim about 0.9 ± 0.4 nm higher than the surrounding intact bilayer (quantified by image segmentation). Two distinct rim types were thus evident: outer patch rims, delineating the patch perimeters, and inner hole rims, encircling the peptide-induced holes. Despite their different locations, both rim types represent exposed bilayer edges of similar physical nature.

At 2 µM Pep-1, the disruptive effects intensified. The holes expanded, and both the inner and outer rims broadened, while their heights remained similar (1.0 ± 0.4 nm). At 4 µM Pep-1, a distinct mode of bilayer disruption emerged. In addition to deep holes and elevated rims, numerous small, bright particle-like features appeared, predominantly co-localized with both rim types. These particles further enhanced the height contrast and prominence of the boundary regions.

The formation of holes likely initiates through preferential binding of Pep-1 to preexisting bilayer defects such as interior pits. The presence of anionic POPS enhances electrostatic attraction between the bilayer surface and the cationic residues of Pep-1, promoting peptide accumulation at these defect sites. Once bound, partial hydrophobic insertion of the peptide perturbs the local packing order and generates lateral stress within the surrounding lipids. This combined electrostatic anchoring and amphipathic insertion destabilize the defect periphery, leading to localized lipid extraction and progressive expansion of the initial pits into larger holes.

The elevated rims—both around the holes and along the outer patch boundaries—represent regions where Pep-1 accumulates and co-assembles with lipids. These rims likely correspond to interfacial zones of altered lipid packing, where peptide binding perturbs the local bilayer organization and stabilizes mixed peptide–lipid assemblies. The rim broadening observed at 2 µM Pep-1 reflects the lateral growth of these complexes as more peptides become available, effectively expanding the extent of the peptide-enriched boundary regions.

At 4 µM Pep-1, numerous particle-like structures appear along both rim types, marking a later stage of membrane remodeling. These particles likely correspond to ordered peptide–lipid aggregates formed when the local membrane surface becomes saturated with bound peptide. Their preferential localization along the rims suggests that the peptide-rich boundaries serve as nucleation sites for secondary aggregation. The formation of such aggregates may further perturb lipid organization by locally dehydrating the interface, thereby amplifying membrane stress and promoting additional lipid rearrangement.

Previous studies have established that negatively charged lipids promote CPP-induced membrane translocation and disruption by providing electrostatic anchoring sites and increasing bilayer flexibility [43,44]. Consistent with this, incorporation of anionic POPS markedly modifies the pathway of Pep-1–induced membrane remodeling compared to zwitterionic POPC bilayers (Figure 1 and Figure 2). In POPC patches, Pep-1 preferentially targets the exposed boundaries, where hydrophobic insertion and lipid extraction progressively erode the patch edges. In POPC/POPS bilayers, a similar lipid extraction process occurs but is spatially redistributed, giving rise to holes surrounded by elevated annular rims as well as rims along the outer patch boundaries. The presence of negatively charged POPS enhances electrostatic attraction with Pep-1, promoting stronger surface binding and localized peptide accumulation. The elevated rims likely represent regions where bound peptide reorganizes lipid packing and co-assembles with lipids into peptide–lipid complexes, with the increased height reflecting compositional rearrangement. At higher peptide concentrations, the formation of rim-associated particles likely reflects the transition from two-dimensional peptide–lipid complexes to three-dimensional aggregates, stabilizing peptide-rich zones and facilitating continued lipid extraction. Thus, the inclusion of POPS stabilizes interfacial peptide–lipid assemblies both at patch boundaries and within the interiors, resulting in a more localized and structurally organized mode of membrane remodeling.

### 3.3. Effect of Cholesterol

We also examined the effect of Pep-1 on POPC + 30 mol% cholesterol bilayer patches (Figure 3). The untreated bilayer displayed smooth patches with well-defined boundaries. Upon addition of 1 µM Pep-1, small, weakly elevated particles appeared across the bilayer surfaces. These faintly bright features were distributed relatively uniformly and likely represent initial peptide binding and insertion sites, where minor lipid rearrangements and curvature perturbations begin to develop.

At 2 µM Pep-1, the small surface particles largely disappeared, and elevated rims became visible along the patch boundaries. These rims were less pronounced than those observed in POPC/POPS patches (Figure 2) and were absent in POPC patches (Figure 1). Their appearance suggests moderate accumulation of Pep-1 at the bilayer edges, where lipid packing is relatively loose and curvature generation is energetically favorable. Peptide association at these sites likely induces subtle bending and local lipid rearrangement, giving rise to boundary rims that mark the early stages of peptide-driven bilayer modulation.

At 4 µM Pep-1, the POPC/cholesterol patches exhibited extensive morphological transformation, with numerous elevated features appearing along both the boundaries and interiors of the patches. High-resolution imaging (Figure 4) revealed these features as continuous, curvilinear ridges or filamentous protrusions that traversed the patch interiors and often connected to the outer rims. Many of these ridges appeared as interconnected, wave-like structures, forming a network of curved elevations across the bilayer surface.

Cholesterol tends to stabilize membranes and diminish CPP-induced disruption [15,45,46,47]. In contrast, the emergence of curved ridges in the present study indicates that Pep-1 actively induces membrane curvature in the presence of cholesterol. Cholesterol enhances lipid packing order while increasing the bilayer’s resistance to compression [48,49,50,51], allowing curvature stress to accumulate as Pep-1 inserts into the membrane. The amphipathic structure of Pep-1 drives its insertion into the upper leaflet through the hydrophobic tryptophan-rich domain, while the cationic residues anchor the peptide electrostatically at the membrane surface. In cholesterol-containing POPC patches, the increased lipid order and rigidity introduced by cholesterol modulate how the bilayer reorganizes in response to peptide insertion. Differences in the interaction dynamics of POPC and cholesterol with Pep-1 likely lead to lipid redistribution, where peptide-complexed lipids segregate into elevated ridges. These protruded ridge-like assemblies arise from cooperative peptide–lipid reorganization and represent regions of concentrated curvature stress. Their formation marks a transition from surface adsorption to active membrane remodeling, in which Pep-1 and cholesterol jointly stabilize locally curved interfaces.

### 3.4. Boundary Effects: Patches vs. Continuous Bilayers

To complement the patch experiments, a continuous bilayer composed of POPC + 30 mol% POPS was examined to assess how Pep-1 interacts with membranes lacking free boundaries (Figure 5). The untreated bilayer formed a smooth, homogeneous film that fully covered the mica substrate, with a few defects.

Upon addition of 1 µM Pep-1, numerous elevated features appeared across the bilayer surface (Figure 5B). These structures exhibited irregular, flower-like shapes and were about 1.0 ± 0.3 nm higher than the surrounding bilayer. The elevated regions represent sites of strong peptide binding and local peptide–lipid co-assembly, driven primarily by electrostatic attraction between the cationic residues of Pep-1 and the anionic headgroups of POPS. Local peptide binding and partial insertion perturb lipid packing, resulting in peptide-rich domains that are thicker than the surrounding membrane. The heterogeneous, flower-like morphology suggests that Pep-1 association begins at discrete nucleation sites and expands radially through lateral diffusion and cooperative lipid reorganization.

When the Pep-1 concentration was increased to 2 µM (Figure 5C), the discrete elevated domains enlarged and merged into an interconnected network of bright regions with a height of 1.6 ± 0.9 nm. This morphological evolution indicates coalescence of peptide-rich domains as local surface coverage increases, transforming the initially uniform bilayer into a laterally phase-separated structure. Interestingly, closer inspection of the boundaries of these elevated regions reveals protrusions with heights exceeding the plateau itself. Such boundary features are inconsistent with a homogeneous peptide overlayer and instead support the formation of a mixed peptide–lipid domain with nonuniform topography. The interconnected high-topography network reflects a peptide-enriched, thickened phase that occupies much of the membrane surface, while the lower areas correspond to less-perturbed bilayer regions. Interestingly, similar bilayer morphological transitions, including domain separation and membrane thickening, have been observed for the CPP penetratin interacting with lipid bilayers [52], suggesting that Pep-1 and penetratin may utilize comparable pathways for peptide-induced membrane remodeling.

Although both the POPC/POPS patches (Figure 2) and continuous bilayers (Figure 5) develop elevated features upon exposure to Pep-1, their morphologies differ markedly due to contrasting mechanical boundary conditions. In bilayer patches, Pep-1 preferentially accumulates along the exposed edges, where lipid packing is less constrained and the boundary can expand to relieve local stress. This results in the formation of pronounced rims outlining the patch perimeters. In contrast, the continuous bilayer is laterally confined by its adhesion to the substrate and lacks free edges that can relax in-plane strain. Under these constraints, stress generated by peptide binding and insertion is accommodated vertically, leading to out-of-plane deformation in the form of discrete, flower-like protrusions. These elevated domains represent regions of peptide-induced thickening and curvature generation within the otherwise constrained membrane.

Together, these observations indicate that Pep-1 remodels membranes through a common mechanism driven by combined electrostatic and hydrophobic interactions. Electrostatic attraction promotes peptide binding to anionic lipid regions, while hydrophobic residues facilitate insertion into the bilayer hydrophobic core. The specific manifestation of this process depends strongly on bilayer geometry and mechanical constraints: boundary-directed rim formation dominates in patches with free edges, whereas localized curvature induction and domain thickening prevail in continuous membranes where lateral expansion is restricted.

### 3.5. Roles of Hydrophobic and Electrostatic Domains in Pep-1–Induced Remodeling

Although our AFM measurements do not provide residue-level resolution, the composition-dependent remodeling behaviors we observe are consistent with the established biophysical roles of Pep-1’s two domains. The tryptophan-rich hydrophobic segment contains bulky indole rings that preferentially partition into the glycerol–carbonyl region of lipid bilayers, promoting partial insertion, interfacial anchoring, and local packing disruption. These properties make tryptophan residues particularly effective at destabilizing exposed or weakly packed patch boundaries, helping to drive the edge erosion observed in POPC patches.

In contrast, the lysine-rich cationic domain mediates strong electrostatic anchoring to anionic membranes. In POPC/POPS continuous bilayers, where no free edges are available, this anchoring likely favors surface-bound accumulation rather than deep insertion. Such surface enrichment can nucleate locally thickened, peptide-rich domains that expand into the flower-like protrusions observed in our AFM images.

Taken together, these considerations suggest that the hydrophobic and electrostatic segments of Pep-1 play complementary roles in membrane remodeling: the tryptophan-rich region supports insertion-driven boundary destabilization, whereas the lysine-rich region promotes surface accumulation and curvature-generating protrusions.

## 4. Conclusions

Pep-1 interacts with lipid bilayers through a combination of electrostatic and hydrophobic forces, but the resulting mode of membrane remodeling is strongly dictated by both lipid composition and bilayer geometry. Figure 6 summarizes the four Pep-1 interaction regimes examined in this study. In POPC patches, Pep-1 preferentially targets exposed patch edges, where hydrophobic insertion and lipid extraction lead to detergent-like lysis and progressive patch disintegration. In POPC/POPS patches, the presence of anionic POPS enhances electrostatic binding, giving rise to elevated rims and particle-like aggregates. In cholesterol-containing patches, Pep-1 generates protruded, ridge-like structures, consistent with curvature induction. In POPC/POPS continuous bilayers, where no free edges are available, Pep-1 forms flower-like protrusions that expand and merge into an interconnected network, reflecting composition-dependent remodeling under constrained, edge-free conditions. Together, these regimes demonstrate that Pep-1 elicits distinct nanoscale responses depending on membrane electrostatics, packing order, and geometric constraints.

These nanoscale morphological signatures can be interpreted within established physical frameworks for membrane remodeling. Local peptide accumulation at bilayer edges can reduce effective line tension [53] and stabilize curved boundaries, giving rise to annular rims. Partial insertion of Pep-1 into one leaflet generates local area asymmetry and curvature stress, which can promote protrusion formation—analogous to membrane budding processes [54]. The lateral merging of elevated peptide-rich regions is consistent with interfacial energy minimization, a process commonly observed in line-energy-driven coarsening of membrane domains [55]. These considerations provide a mechanistic foundation that links the observed AFM features to general principles of curvature elasticity and boundary energetics.

It is important to recognize that the reported membrane remodeling is observed in supported lipid bilayers, where adhesion to the mica substrate substantially restricts the membrane’s ability to relax [41]. As a result, Pep-1–induced perturbations appear static and cumulative: once a defect forms, the bilayer cannot recover, and subsequent peptide additions amplify the disruption. In contrast, cellular membranes are fluid, deformable, and unconstrained, allowing peptide-induced perturbations to be transient rather than persistent. Thus, the AFM images should not be interpreted as evidence that Pep-1 acts solely as a membrane-disrupting peptide, but rather as revealing structural modes that would typically represent short-lived intermediate states during its interaction with living membranes.

Our observations highlight that Pep-1 does not follow a single, unified mechanism of action but instead adapts its interaction mode to the membrane’s compositional and mechanical context. Such flexibility—ranging from boundary erosion to curvature induction—demonstrates the peptide’s capacity to remodel membranes through distinct pathways. The ability of Pep-1 to deform, reorganize, and partially destabilize lipid bilayers in a controlled and self-limiting manner supports the view that these perturbations facilitate transient structural rearrangements consistent with proposed direct translocation mechanisms of cell-penetrating peptides. Moreover, the composition-dependent remodeling pathways identified here may inform the rational design of next-generation delivery vectors: preferential curvature generation suggests strategies for enhancing uptake in curvature-rich cellular regions, while the strong dependence on anionic lipid content underscores the potential to target or exploit negatively charged membrane domains. By tuning peptide charge distribution, hydrophobicity, or domain specificity, these mechanistic insights could help optimize CPP-based drug delivery systems for improved cellular targeting and translocation efficiency.

## Figures and Tables

**Figure 1 membranes-15-00373-f001:**
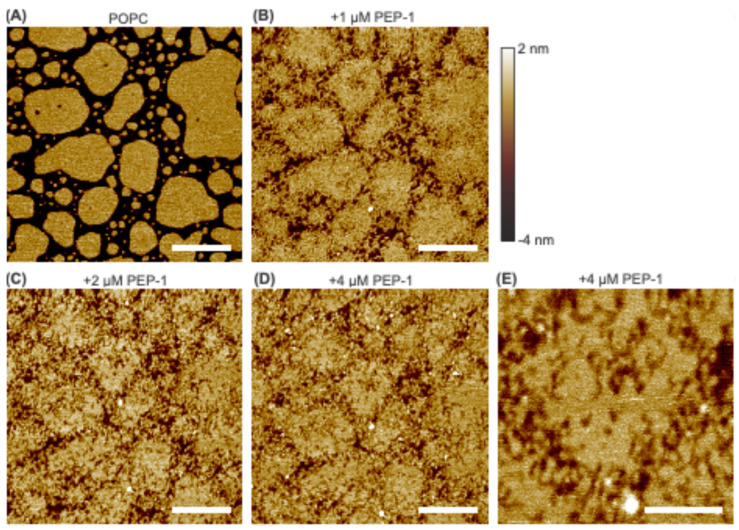
AFM height images of POPC bilayer patches exposed to increasing concentrations of Pep-1: 0 µM (**A**), 1 µM (**B**), 2 µM (**C**), and 4 µM (**D**). A high-resolution image acquired at 4 µM Pep-1 is shown in (**E**). The height scale is indicated by the color bar. The same color scheme is used throughout the paper. White scale bars represent 500 nm in (**A**–**D**) and 200 nm in (**E**).

**Figure 2 membranes-15-00373-f002:**
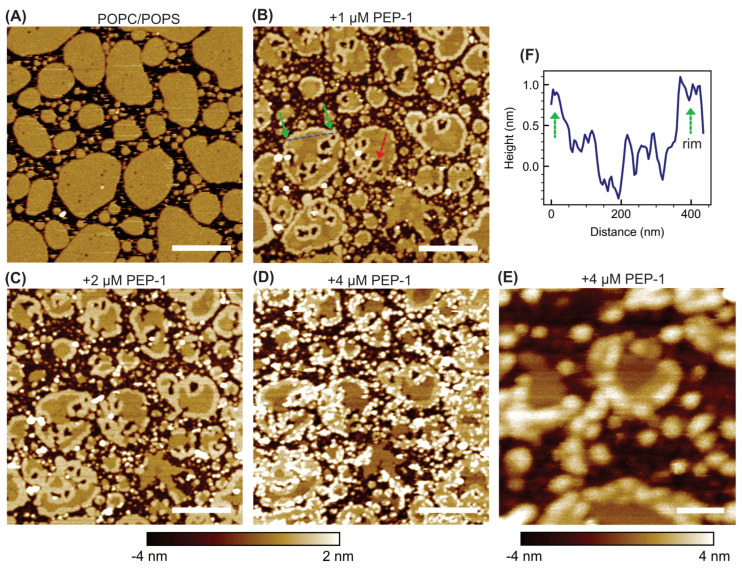
AFM height images of POPC + 30 mol% POPS bilayer patches exposed to increasing concentrations of Pep-1: 0 µM (**A**), 1 µM (**B**), 2 µM (**C**), and 4 µM (**D**). A high-resolution scan (500 nm × 500 nm) acquired at 4 µM Pep-1 is shown in (**E**). The blue dashed line in (**B**) indicates the trajectory used to obtain the height profile across a bilayer patch shown in (**F**). The green arrow in (**B**) highlights elevated boundary rim, while the red arrow marks hole formation within the patch interior. The height scale for panels (**A**–**D**) ranges from −4 nm to 2 nm; for panel (**E**), it ranges from −4 nm to 4 nm. Scale bars: 500 nm for (**A**–**D**) and 100 nm for (**E**).

**Figure 3 membranes-15-00373-f003:**
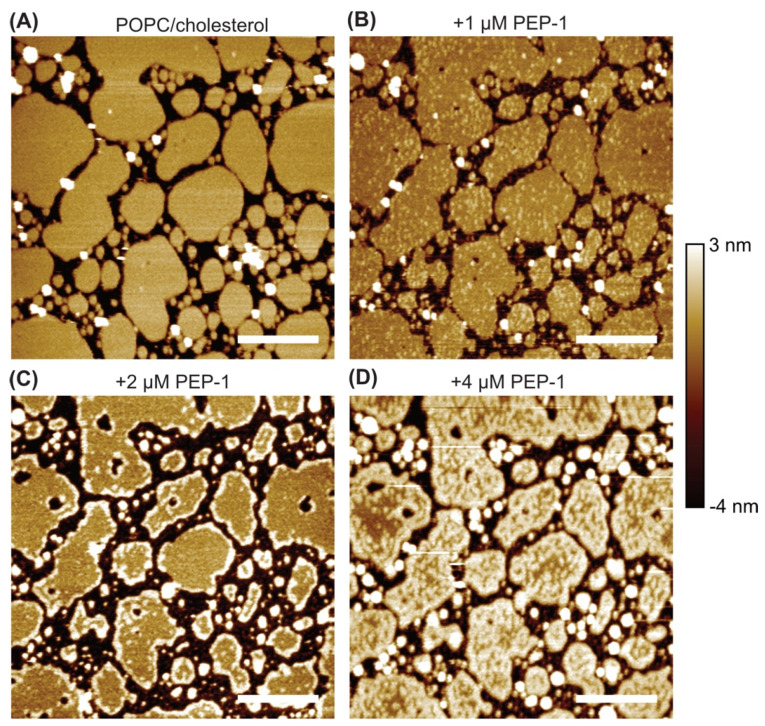
AFM height images of POPC + 30 mol% cholesterol bilayer patches exposed to increasing concentrations of Pep-1: 0 µM (**A**), 1 µM (**B**), 2 µM (**C**), and 4 µM (**D**). Scale bars: 500 nm.

**Figure 4 membranes-15-00373-f004:**
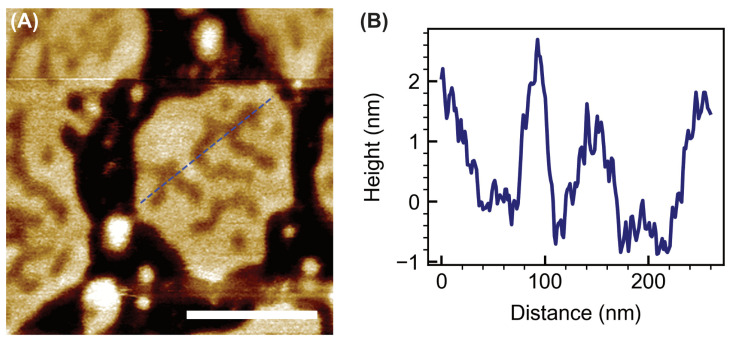
High-resolution AFM image of POPC + 30 mol% cholesterol bilayer patches treated with 4 µM Pep-1 (**A**). Scale bar: 200 nm. The blue dashed line in (**A**) indicates the trajectory used to obtain the height profile shown in (**B**).

**Figure 5 membranes-15-00373-f005:**
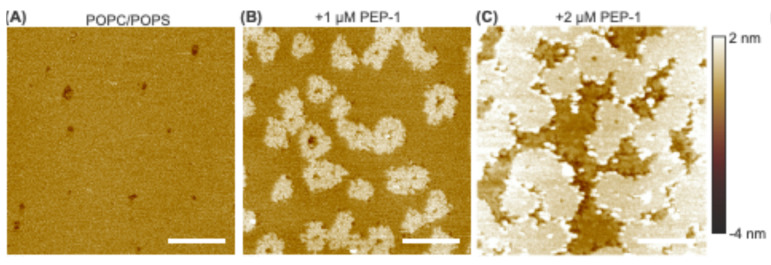
AFM height images of a continuous POPC + 30 mol% POPS bilayer exposed to increasing concentrations of Pep-1: 0 µM (**A**), 1 µM (**B**), and 2 µM (**C**). Images were acquired from neighboring regions of the same bilayer following sequential peptide additions. Scale bars: 500 nm.

**Figure 6 membranes-15-00373-f006:**
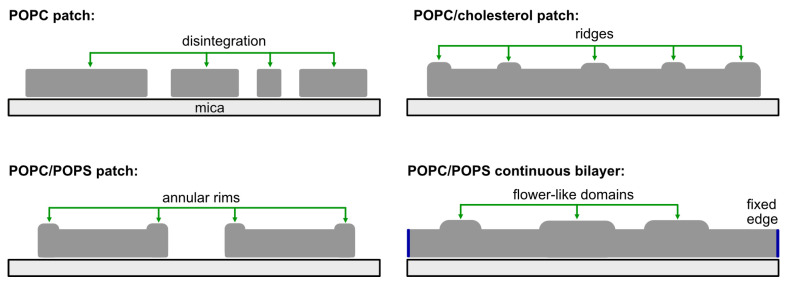
Schematic summary of Pep-1–induced membrane remodeling. POPC patch: Pep-1 targets patch edges, causing lipid extraction, edge erosion, and detergent-like disintegration. POPC/POPS patch: Strong electrostatic binding leads to elevated rims. POPC/cholesterol patch: Pep-1 induces ridge-like protrusions driven by curvature generation. POPC/POPS continuous bilayer: In the absence of free edges, Pep-1 forms flower-like protrusions that can grow into an interconnected network.

## Data Availability

The data presented in this paper are available upon request.

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
