# Peer review of "Curvature-Induced Membrane Remodeling by the Cell-Penetrating Peptide Pep-1"

_membranes, 2025, doi:10.3390/membranes15120373_

Round 1

Reviewer 1 Report

Comments and Suggestions for Authors

This manuscript presents a study using atomic force microscopy to investigate how the cell-penetrating peptide Pep-1 interacts with and remodels lipid membranes. The work revealed that the specific mechanism of membrane disruption, including destabilisation and curvature generation, is highly dependent on the lipid composition and mechanical properties of the bilayer. The study offers new insights into Pep-1-mediated membrane remodelling and is well-presented, with an introduction supporting its background. However, minor revisions are necessary before it can be published.

Below are my comments aiming to improve this manuscript before being accepted for publication.

1) The manuscript provides a detailed description of the nanoscale defects and roughened texture within the patch interiors upon Pep-1 interaction, starting around line 173. However, the corresponding AFM images (Figure 1) primarily showcase the overall patch disintegration and edge erosion. Could the authors provide higher-magnification images focused specifically on the patch interiors to visually support the described phenomena?

2) The reference list is heavily weighted towards foundational and older studies (e.g., many from 1988-2013). While these are crucial for establishing the field, the manuscript would benefit from a more balanced integration of recent literature (e.g., from the last 5 years) on cell-penetrating peptide mechanisms and membrane remodelling. Could the authors incorporate more contemporary references to better situate their findings within the current state of the art?

3) The conclusions effectively summarise the key biophysical findings but could be enhanced by a more detailed discussion of the potential biological or practical implications. Could the authors elaborate further on how these composition-dependent remodelling pathways might influence the design of more efficient peptide-based drug delivery vectors? For instance, how could the insights on membrane curvature generation or anionic lipid dependency be leveraged to improve cellular targeting or translocation efficiency in therapeutic applications?

4) The study is predominantly qualitative, relying on representative AFM images to demonstrate morphological changes. To support the conclusions, could the authors provide quantitative analyses? For example, could they measure and report the changes in surface roughness, the distribution of hole diameters in POPS-containing bilayers, or the frequency and height of the ridge-like features in cholesterol-containing membranes across multiple independent experiments? Statistical analysis would help substantiate the claims about the extent and significance of the observed remodelling.

5) The proposed mechanism appropriately highlights the roles of electrostatic and hydrophobic forces. However, given the peptide's distinct domains (lysine-rich vs. tryptophan-rich), could the authors speculate more deeply on how each domain might contribute to the different remodelling outcomes? For example, how might the specific role of tryptophan residues in hydrophobic insertion versus lysine residues in electrostatic anchoring differentially drive edge erosion versus the formation of flower-like protrusions?

Reviewer 2 Report

Comments and Suggestions for Authors

The evidence presented in this work indicates that PEP1 is a membrane-disrupting peptide rather than a CCP. CPPs are expected to enter cells without damaging the membrane. Please analyze this point.

Are the white regions in the images saturated regions (thicker than the maximum height measured by the instrument) or are they an artifact?

Regarding Figure 5, these results can be interpreted as the appearance of a new phase, and therefore, the peptide induces phase segregation, as described by the authors. However, the images can also be explained by considering that the peptide adsorbs onto the bilayer, forming a peptide layer on the membrane. I think the experiments don't allow us to distinguish between these two situations, do they?

In general, the image descriptions are qualitative, except for the high profiles in some images. It would be useful to have quantitative (and comparative) data on the high profiles, including statistics. It would also be useful to quantify the size of the regions with each thickness.

In some cases, the thicker regions are attributed to peptide accumulation. For example: "The elevated regions represent sites of strong peptide accumulation, driven primarily by electrostatic attraction between the cationic residues of Pep-1 and the anionic headgroups of POPS” (lines 303 - 305).

In other cases, the thicker regions are attributed to the formation of peptide-lipid complexes or an increase in membrane thickness. For example: "The elevated rims—both around the holes and along the outer patch boundaries—represent regions where Pep-1 accumulates and co-assembles with lipids." (lines 207-208).

How can you differentiate between these two situations?

Please include the peptide sequence, indicating the charged and hydrophobic regions.

Reviewer 3 Report

Comments and Suggestions for Authors

The manuscript by Y. I. Gamage and J. J. Pan presents a detailed AFM investigation of how the cell-penetrating peptide Pep-1 remodels supported lipid bilayers of different compositions (POPC, POPC/POPS, and POPC/cholesterol) and geometries (patches vs. continuous films). The work provides valuable nanoscale evidence of composition-dependent membrane perturbations, revealing curvature-driven reorganization and structural remodeling. The data are well presented, and the topic is relevant to membrane biophysics and peptide–membrane interaction studies. However, several points should be addressed before the manuscript can be considered suitable, from my side, for publication in Membranes. My detailed comments are listed below.

- While the Introduction provides an adequate overview of cell-penetrating peptides (CPPs), it would benefit from a broader contextualization that also includes antimicrobial peptides (AMPs). Although these two classes of biomolecules are often discussed separately, several studies—both experimental and theoretical—have highlighted striking similarities in their mechanisms of membrane interaction. In particular, some AMPs, including clinically relevant lipopeptides such as daptomycin, exhibit insertion and membrane-remodeling behavior comparable to that of CPPs, suggesting that they may share common physicochemical driving forces. It would therefore be valuable to mention experimental investigations on such systems (e.g., Eur Biophys J 49, 401–408 (2020), https://doi.org/10.1007/s00249-020-01445-w which have elucidated peptide–membrane perturbation mechanisms using GUVs and SLBs mimicking biological membranes.

- Since the manuscript discusses nanoscale peptide–membrane interactions, the Introduction would also benefit from acknowledging a broader mechanistic perspective that has emerged in recent years. For instance, several studies have pointed out notable analogies between peptide-induced membrane perturbations and those produced by functionalized metallic nanoparticles, particularly amphiphilic gold nanostructures. These parallels have opened new perspectives in the design of bioinspired nanomaterials and in the mechanistic interpretation of peptide–lipid interactions at the nanoscale. Including a brief discussion of this conceptual connection would broaden the scope of the Introduction and provide a richer mechanistic framework for the presented work.

- While the AFM results are comprehensive, the authors could better connect their findings to physical models of peptide-induced curvature and lipid reorganization. A short discussion linking observed protrusions or rims to membrane tension or line-energy minimization would strengthen mechanistic insight.

- In lines 153–160, the authors describe a “detergent-like lysis mechanism.” It is not fully clear whether this mechanistic description is specifically referring to Pep-1, or whether it represents a more general behavior observed in structurally related peptides. If this section is meant to describe Pep-1, appropriate references should be added to support the statement. Conversely, if the authors intend this as a speculative or generalized mechanistic analogy, this should be explicitly stated to avoid ambiguity.

- At line 225 authors state: “The formation of such aggregates may further perturb lipid organization by locally stiffening or dehydrating the interface”. This is an interesting and plausible hypothesis; however, it would be helpful to clarify whether this statement is supported by any experimental evidence. For instance, have the authors performed (or found in literature) any nanoscale mechanical measurements—such as AFM force spectroscopy, indentation analysis, or related techniques—that could corroborate local stiffening or dehydration effects? I suggest mentioning any available studies or citing related work in which similar nanoscale effects were observed upon interaction with other antimicrobial peptides, as this would help contextualize the proposed mechanism.

- In Section 3.4 Boundary Effects: Patches vs Continuous Bilayers, the discussion would benefit from a clearer indication of the experimental temperature conditions. Since patch stability, defect evolution, and peptide-induced reorganization can strongly depend on the thermodynamic state of the lipid bilayer, temperature control is a key parameter in this type of analysis. Even small variations around the lipid phase transition may significantly alter boundary behavior and mechanical response. I therefore recommend specifying the exact temperature at which the AFM measurements were performed, and—if applicable—commenting on whether the system was investigated relative to the phase transition of the lipid mixture. This information would help readers better interpret the observed nanoscale organization and its underlying thermodynamic basis.

Minor

- From a graphical perspective, I would suggest ensuring a consistent presentation style across figures. In particular, if possible, Figure 2 would benefit from a high-resolution panel (or a zoom-in view) similar to the one provided in Figure 1. This would improve visual clarity and allow the reader to better appreciate the nanoscale morphological details discussed in the text.

- Consider including a schematic or model summarizing the three interaction regimes (POPC, POPC/POPS, and POPC/Chol), which would visually unify the proposed mechanisms.

- Line 98: facilitate instead of facilitates

Round 2

Reviewer 2 Report

Comments and Suggestions for Authors

My comments were answered

Reviewer 3 Report

Comments and Suggestions for Authors

The authors addressed all the points raised up in my first revision round.